# Exploring the Provider-Level Socio-Demographic Determinants of Diet Quality of Preschool-Aged Children Attending Family Childcare Homes

**DOI:** 10.3390/nu12051368

**Published:** 2020-05-11

**Authors:** Alison Tovar, Patricia Markham Risica, Andrea Ramirez, Noereem Mena, Ingrid E. Lofgren, Kristen Cooksey Stowers, Kim M. Gans

**Affiliations:** 1Department of Nutrition and Food Sciences, University of Rhode Island, Kingston, RI 02881, USA; andrea_ramirez@uri.edu (A.R.); alison.tovar@gmail.com (N.M.); ingrid_lofgren@uri.edu (I.E.L.); 2Center for Health Promotion and Health Equity, Brown University, South Main Street, Providence, RI 02912, USA; patricia_risica@brown.edu; 3Department of Allied Health Sciences, University of Connecticut, 358 Mansfield Rd, Storrs, CT 06269, USA; kristen.cooksey@uconn.edu; 4Human Development & Family Sciences, 348 Mansfield Road, Unit 1058, University of Connecticut, Storrs, CT 06269, USA; kim.gans@uconn.edu

**Keywords:** diet quality, childcare, healthy eating index

## Abstract

Since food preferences develop during early childhood and contribute to dietary patterns that can track into adulthood, it is critical to support healthy food environments in places where children spend significant amounts of time in, such as childcare. It is important to understand what factors influence the diet quality of children cared for in family childcare homes (FCCH). Methods: This study used baseline data from a cluster-randomized trial in FCCH, Healthy Start/Comienzos Sanos. Surveys capturing providers’ socio-demographic characteristics were completed. Food and beverage consumptions were estimated using the Dietary Observation in Childcare protocol, and diet quality was calculated using the Healthy Eating Index (HEI)-2015. Comparison of mean HEI scores by provider socio-demographic variables were completed using ANOVAs, followed by multiple linear regression models for significant variables. Post-hoc ANOVA models compared mean HEI-2015 sub-components by income and ethnicity. Results: Significant differences in mean HEI-2015 scores were found for provider income level (less than $25,000, HEI: 64.8 vs. $25,001–$50,000: 62.9 vs. $75,001 or more: 56.2; *p* = 0.03), ethnicity (Non-Latinx: 56.6 vs. Latinx: 64.4; *p* = 0.002), language spoken outside of childcare (English: 58.6 vs. Spanish: 64.3, *p* = 0.005), and language spoken in childcare (English: 59.6 vs. Spanish: 64.4; *p* = 0.02). In linear regression models, a higher provider income ($75,001 or more) was negatively and significantly associated with the total HEI-2015 scores (β = −9.8, SE = 3.7; *p* = 0.009) vs. lower income (less than $25,000). When entering provider income and ethnicity to the same model, adjusting for Child and Adult Food Program (CACFP), only ethnicity was significant, with Latinx being positively associated with total HEI-2015 scores vs. non-Latinx (β = 6.5, SE = 2.4; *p* = 0.007). Statistically significant differences were found by ethnicity and language for greens/beans, total protein, and seafood and plant protein HEI-2015 component scores. Discussion: Lower income, and Latinx providers cared-for children had higher diet quality in FCCH compared to the other providers. Future studies should better understand what specific foods contribute to each of the HEI-2015 components in order to better tailor trainings and interventions.

## 1. Introduction

Suboptimal diet is the leading risk factor for death and disability in the US [1,2]. The diet quality of U.S. children is generally poor, with an inadequate intake of fruits, vegetables, and whole grains, and over-consumption of energy-dense snacks and beverages [3,4,5,6], especially among low-income and ethnic minorities [7,8]. Poor diets contribute substantially to increased risk of chronic conditions including obesity, cardiovascular disease, specific cancers, and diabetes [9,10,11,12]. Since food preferences develop during early childhood [13,14] and contribute to dietary patterns that can track into adulthood [15], it is critical to support healthy food environments in places where children spend their time.

The childcare environment is an ideal setting to promote children’s eating behaviors, given that 60% of US children under five years of age receive non-parental care [16]. Children under five spend on average over 27 h a week in childcare settings [17], and it is recommended that they obtain 1/2 to 2/3 of their daily nutrients from meals and snacks served in childcare [18,19,20]. Furthermore, while the home environment has a considerable influence on children’s dietary habits [21,22], licensed childcare settings are subject to nutritional regulations and policies and are supported by federal programs, such as the Child and Adult Food Program (CACFP) [23]. 

Most of the studies exploring what children eat in childcare, have been conducted in childcare centers [24,25,26,27,28,29,30,31,32,33,34,35]. These studies found that there is room for improvement of their diet quality, in particular with regards to consuming high amounts of saturated fat, sodium, and added sugars [25,26,28,36]. However, while three million US children are enrolled in family childcare homes (FCCH) [37]—a form of childcare operated from the home of a non-relative—studies of children’s eating in FCCH are limited. Of the few studies, like in childcare centers, there appears to be room for improvement in children’s diets in FCCH [34,38,39,40]. More studies are needed to examine the diet quality of children cared for in FCCH and determine the factors that impact this diet quality. 

In the home setting, demographic factors such as parent/caregiver age [41], education [42], and race [43,44] have been associated with children’s diet quality. Acculturation also impacts children’s diet quality [44]. It is well known that the healthfulness of the traditional Latinx diet, which is high in legumes and fruit [45,46], tends to deteriorate with the acculturation process due to changes in both the environment and social factors. It is plausible to expect that similar factors such as provider age, education, income, and acculturation may also play a role in diet quality specific to FCCH.

Statewide survey research with FCCH providers found that provider ethnicity was related to reporting certain nutrition-related best practices [47]. Furthermore, focus groups with Latinx and non-Latinx FCCH providers also indicate the potential influence of culture on foods served to children in their care. [48,49,50] The relationship between acculturation, culture, and ethnicity is complex and multifaceted; [51] therefore, examining the FCCH provider ethnicity and proxy measures of acculturation such as language preference [52] as possible predictors of child diet quality could inform contextual-based strategies to maintain or improve healthy eating environments in FCCH. Therefore, the aim of this study was to explore the diet quality of meals and snacks consumed by preschool-aged children (i.e., ages 2 to 5) in FCCH and explore whether diet quality differs by sociodemographic characteristics and acculturation of the FCCH providers.

## 2. Methods

### 2.1. Study Design

This study used baseline data from a cluster-randomized trial, Healthy Start/Comienzos Sanos, which evaluated the efficacy of a multicomponent intervention to improve the food, physical activity, and screen time environments of FCCH, as well as the diet and physical activity of the 2-to-5-year-old children in their care [53]. Details about the study recruitment, intervention, and evaluation are discussed elsewhere [53], but the methods relevant to the current analyses are described below. The baseline data collection was conducted from January 2016 until July 2018. The Institutional Review Boards of Brown University, University of Rhode Island and University of Connecticut, approved all the study procedures and materials. Participating Family Child Care Home Providers (FCCP) and the parents of children in FCCH provided informed consent. The study was registered at ClinicalTrials.gov with the identifier NCT02452645.

### 2.2. Setting, Participants, and Eligibility Requirements

The study enrolled and measured 374 children attending 119 FCCH within 60 miles of Providence, Rhode Island (RI). To be eligible, the FCCH needed to be in operation for at least 6 months. The FCCH provider had to read and speak Spanish and/or English and care for at least one unrelated 2-to 5-year-old child for at least 10 h per week, who ate at least one meal and snack per day at the FCCH. Eligible providers completed a 30-min baseline telephone survey followed by a 30-min in-person survey at the FCCH. Once the parents of eligible 2-to-5-year-old children at the FCCH consented, a two-day observation was scheduled. The providers received $25 for completing the baseline survey and $50 for the two-day baseline observation.

### 2.3. Measures

#### 2.3.1. Demographics

Providers reported their gender, ethnicity, and race on the telephone survey and the following variables on the in-person survey: age, household income, marital status, education, years as a childcare professional, number of children currently in their care (and how many are their own children or grandchildren), the average number of hours that children spend daily at the FCCH, and whether the FCCH accepted Child and Adult Care Food Program (CACFP) benefits. The following proxy measures of acculturation were asked on the survey: years in the US, country of origin, language spoken in their home outside of childcare, and languages spoken with the children in their FCCH (response options included: English only; Spanish only; both—more English than Spanish; both—equal amounts of time; and both—more Spanish than English).

#### 2.3.2. FCCH Observation

Trained researchers visited the FCCH for two full days to observe the FCCH environment, including the meals and snacks that the children consumed. On visit days, the research team arrived before the children had their first meal at the FCCH. The observers remained at the FCCH for as long as the children were there (except at nap time) at a convenient location in the FCCH so as not to interfere with their routine. Each observer examined a maximum of three children; if there were more than three children to observe, two observers conducted the FCCH observation.

The children’s food and beverage consumptions were calculated/estimated using the Dietary Observation in Childcare (DOCC) protocol [54], a valid and reliable tool for capturing the dietary intake of children in childcare [55]. For this project, trained data collectors observed and recorded all meals and snacks served to participating children across two full days of childcare. Data collectors estimated the quantity of food and beverages served, added (i.e., second helpings), exchanged, wasted, and remaining following the end of each meal and snack in order to calculate the total quantity consumed by each child. Additional details about the recipes, preparation methods, or brand names of the food or beverages served were requested from the FCCP.

The DOCC-based foods consumed, methods of food preparation, and ingredients of any recipes were entered into the Nutrition Data System for Research (NDSR) [56]. The NDSR determined the energy and nutrient content of the foods and beverages consumed per child throughout the day at the FCCH. Then, an average intake over the two observation days was computed. Although most children were present for both days of observations, if a child did not have two days of observation (*n* = 56, 15% of children), a single observed day was used for the analysis instead of the two-day average, as has been done in previous studies [27,28].

The NDSR outputs containing the total nutrient data and food group servings were exported for the calculation of the Healthy Eating Index-2015 (HEI-2015) scores for each child, which were then averaged for each FCCH setting. The HEI is density-based (e.g., amounts per 1000 kcal) rather than absolute amounts, which makes it comparable across individuals and settings [57,58]. As a measure of diet quality, the HEI-2015 scores reflect adherence to the food group components of healthy food patterns described in the 2015–2020 Dietary Guidelines for Americans (DGA) [21]. The DGA-recommended food patterns are based on foods typically consumed by Americans. The HEI-2015 is comprised of 13 food group components divided into adequacy components (total fruit, whole fruit, total vegetable, greens and beans, whole grains, dairy, total protein foods, seafood and plant proteins) and moderation components (fatty acids, refined grains, sodium, added sugars, and saturated fat) [59].

The food group components were derived using established publicly available National Cancer Institute SAS code. Each of the components are scored using a density basis (per 1000 kcal), except fatty acids, which is a ratio of unsaturated to saturated fatty acids, and each component has a standard for a minimum (0 points) or maximum (5 or 10 points) score. The scores for each component were summed to calculate a total score ranging from 0 to 100, with a higher score indicating better adherence to the 2015–2020 Dietary Guidelines. HEI scores > 80 indicate a “good” diet, scores ranging from 51 to 80 reflect a diet that “needs improvement”, and HEI scores < 51 imply a “poor” diet [60]. 

### 2.4. Statistical Analysis

Descriptive statistics, including frequencies and proportions, were calculated for all the variables using SAS Studio 9.4. Continuous variables, such as age and years in the US, were dichotomized at the median. Some categorical variables, after examination of the initial frequencies, were re-categorized into fewer categories (education was collapsed into high school/GED and associate, bachelor or graduate; years in the US was dichotomized at the mean into less than or more than 23 years). We first identified all possible determinants: provider age, race, income, marital status, education, having a child development credential, number of hours worked per week, total number children in their childcare home, ethnicity, years in country, language spoken in their home and in the FCCH, and if they receive CACFP subsidies. Next, we compared mean HEI-2015 scores by the different demographic variables using one-way ANOVAs. Socio-demographic variables that were statistically significantly associated with HEI-2015 (*p* < 0.05) were included in multiple linear regression models. Given that low-income FCCH providers often have children who are eligible to receive CACFP reimbursements, and given that CACFP has been shown to impact the diet quality of foods in childcare [34,61], the models were adjusted for this variable. To better understand what particular components of the HEI-2015 were contributing to the total HEI-2015 differences, post-hoc ANOVA models were created to compare HEI-2015 sub-components by income and ethnicity. Radar plots were created to visualize each component score simultaneously.

## 3. Results

The FCCH providers were all female and were, on average, 48.9 years old (Table 1). A little less than half of respondents were White (42.5%) with the remainder being Black (15%) or other (34.2%). Most of the providers (69.8%) reported an annual household income of $25,001–$50,000 per year and being married or living with a partner (75.0%). Less than half reported having a high school or GED degree (43.3%), while 56.7% reported having an associate, bachelor, or graduate degree. Over half of the providers were born outside of the US (70%) and reported being Latinx (67.5%), with over half identifying as Dominican (61.2%), followed by Colombian (12.4%), and Puerto Rican (8.3%). More than half (54.6%) reported speaking Spanish only or more Spanish than English outside of childcare, with 45.8% speaking Spanish only or more Spanish than English in childcare. Most reported working for more than 50 h a week (72.5%), with six or more children in their care (78.9%) and receiving CACFP subsidies (82.5%). 

The average HEI-2015 score of the children in this sample was 61.8 ± 11.0, indicating that children’s diet in the family child care home setting “needs improvement” [62]. The HEI-2015 scores differed significantly by income, ethnicity, language spoken outside of childcare, and language spoken in childcare (Table 1). 

In univariate linear regression models, a higher provider income ($75,001 or more) was negatively and significantly associated with total HEI-2015 scores (model 1, β = −9.8, SE = 3.7, *p* = 0.009) vs. lower income (less than $25,000) (Table 2). In another separate univariate model (model 2), being cared for by Latinx providers was positively and significantly associated with total child HEI-2015 scores vs. non-Latinx providers (β = 8.0, SE = 2.0, *p* = 0.001). Similarly, in the univariate model assessing language preference (model 3 and 4), providers speaking Spanish or mostly Spanish in the home and in childcare were also positively and significantly associated with the total HEI-2015 scores vs. providers speaking English (β = 5.9, SE = 1.9, *p* = 0.003, (β = 5.1, SE = 2.0, *p* = 0.01, respectively). When entering provider income and ethnicity to the same model (model 5), adjusting for CACFP, only ethnicity was significant, with Latinx being positively associated with total HEI-2015 scores vs. non-Latinx (β = 6.5, SE = 2.4, *p* = 0.007). When adding language (spoken outside of childcare and in childcare) and income together (model 6), adjusting for CACFP, there were no significant differences (Table 2).

The HEI-2105 greens/beans subcomponent score differed by provider income, whereby higher incomes ($75,001 or more) had lower scores (0.6) as compared to FCCP with incomes of $25,001–$50,000 (2.3) and less than $25,000 (2.5, *p* = 0.004) (Table 3). These differences across income categories were also true for the total protein score (2.6 vs. 3.6 and 3.9, respectively, *p* = 0.02) and the seafood and plant protein score (0.9 vs. 2.4 and 3.1, respectively, *p* = 0.004). The same HEI-2015 subcomponents significantly differed by ethnicity, whereby Latinx vs. non-Latinx providers had higher scores for greens/beans (2.7 vs. 0.5, *p* < 0.0001), total protein (3.7 vs. 2.8, *p* = 0.005), and seafood and plant proteins (2.8 vs. 1.1, *p* < 0.0001). In addition, the refined grains component also significantly differed, with Latinx vs. non-Latinx and those that were Latinx having higher scores (5.4 vs. 3.8, *p* = 0.01). Data for the languages spoken in childcare and at home were similar to the Latinx data, therefore data is only presented for Latinx vs. non-Latinx.

To visualize each of the individual sub-components by income and ethnicity, the component scores were plotted as a percentage of its maximum points on one of the axes. These radar graphs illustrate the different patterns of quality by income and ethnicity as described above, whereby greens and beans, total protein foods, and seafood and plant protein differ across income, ethnicity, and language, with refined grains being different across ethnicity and language (Figure 1 and Figure 2). 

## 4. Discussion

The goal of this study was to explore possible provider sociodemographic characteristics associated with the diet quality of meals and snacks consumed by preschool-aged children (2 to 5 years of age) in family childcare homes (FCCH). We found that provider income, ethnicity, and the primary language spoken both outside and within the childcare setting were associated with the total HEI-2015 scores. A higher family childcare provider (FCCP) income level was associated with the lower diet quality of the children they cared for in their FCCH, while the FCCP being Latinx was associated with the higher diet quality of children in their care. The primary HEI-2105 subcomponents driving the differences in diet quality were greens/beans, protein foods, and refined grains.

As most studies report that diet quality increases as income increases [62], our finding that the children cared for by higher-income FCCP had a lower diet quality as compared to lower-income FCCP was unexpected. We considered that lower-income FCCH could be more likely to receive CACFP subsidies and therefore follow stringent CACFP nutrition guidelines; however, in our study, CACFP status was not related to children’s HEI scores or to provider income. It is possible that, because most providers in our study received CACFP subsidies, we did not observe this to be true. To our knowledge, no other studies have explored the income of FCCH providers and how it may impact the diet quality of the foods that children consume in their care. It is possible that higher income FCCP may be more likely to purchase and serve more expensive prepared foods like chicken nuggets, pizza, etc., while lower income FCCP may purchase and serve less expensive foods such as rice, beans, and eggs and/or cook more from scratch. Future studies should explore what specific foods may be contributing to these findings.

We also found that a Latinx ethnicity was associated with a higher diet quality. Very few studies have explored the impact of childcare provider ethnicity on the diet quality of children, yet a statewide survey with FCCH providers found that provider ethnicity was related to reporting certain nutrition best practices [47]. For example, Latinx providers were less likely to report feeding, eating, and drinking unhealthy foods/beverages in front of children and having screens on during meals and more likely to seek nutrition training compared to non-Latinx providers. Furthermore, qualitative research with Latinx and Latino FCCP also indicates the potential influence of culture on the foods served to children in their care [48,49,50]. In one of these qualitative studies, Latinx providers discussed the importance of several positive practices, including becoming familiar with foods served at home, encouraging healthy foods, role modeling, the importance of meal planning, and buying and preparing meals in advance, in addition to participating in nutrition workshops and training [49]. Outside of childcare, nationally representative data [62,63] as well as other studies with children have found that the diet quality scores are highest among Latinx households and lowest in non-Hispanic black participants [64,65,66]. 

We also found that in unadjusted models, language preference, which may be a proxy measure of acculturation, was associated with higher diet quality. Upon adjusting for income and CACFP, this was no longer significant, however. Other studies have found [67]—although not in childcare settings—that a Spanish speaking language preference is associated with a higher diet quality. For example, one study completed among 1298 Latinx youth found that those who were Spanish speaking had higher diet quality scores compared to those who were English speakers [68]. Similarly, in another study—the Hispanic Community Health Study—among 15,633 Latinx adults, Spanish language preference was associated with better adherence to dietary guidelines [69]. Future studies should explore other measures of acculturation beyond proxy measures to better understand the relationship with diet quality in a childcare setting. 

Dietary acculturation refers to the multi-dimensional process in which immigrants adopt, negotiate, and alter food attitudes and beliefs from the dominant culture that can result in changes to dietary consumption [70]. As parents acculturate to the US, their children’s consumption of energy-dense snacks and sweetened beverages increases [43,71]. However, other studies indicate that acculturation can be both protective and a risk factor for healthy dietary habits among Latinx populations, as it is associated with higher consumption of fruits, vegetables, and whole grains, but also the consumption of more energy dense food [72,73,74,75]. Data from the National Health and Nutrition Examination Survey has shown that those with lower acculturation scores had better odds of higher overall diet quality and subcomponent scores [76]. These data are primarily from Mexican American populations, but do suggest similar trends with the Hispanic subgroups in our study. Therefore, lower levels of acculturation could potentially influence the greater consumption of greens/beans and seafood and plant protein foods and the lower consumption of refined grains in FCCH. Furthermore, this may also be driven by the high consumption of beans in this Latinx population [77].

This study was not without limitations, and the results should be interpreted in the context of social and environmental factors. Firstly, our sample was primarily composed of Dominican FCCP with a mean age of 50 years old recruited in the greater Providence, RI area. Given the heterogeneity of Latinx subgroups across the US [69], caution should be taken when generalizing these results. Second, there may other unmeasured confounding factors which we were unable to adjust for. Future studies should be replicated in other settings to validate our findings. In addition, the cross-sectional design of our study limits the ability to draw causal inferences. Although social desirability bias cannot be discounted when interpreting our results, observation is the gold standard by which to assess the dietary consumption of children in childcare [54,78,79]. 

## 5. Conclusions

In conclusion, we found that family childcare home provider income, ethnicity, and language were important contributing factors to the diet quality of children in family childcare homes. These findings suggest that culture and food preferences are important contributors in this childcare setting. In addition, these findings highlight the need for culturally appropriate trainings and interventions that are better tailored for both non-Latinx providers and those of both lower and higher incomes. Future studies should further explore the food items contributing to these income and ethnic differences in diet quality and other associated socio-cultural contextual factors in order to better tailor trainings and interventions.

## Figures and Tables

**Figure 1 nutrients-12-01368-f001:**
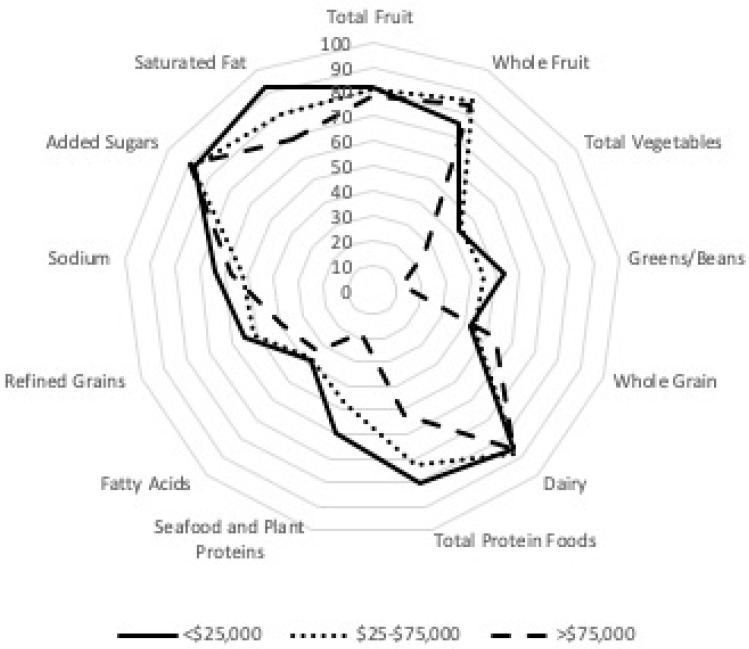
Radar chart of HEI-2015 for 119 FCCH by income.

**Figure 2 nutrients-12-01368-f002:**
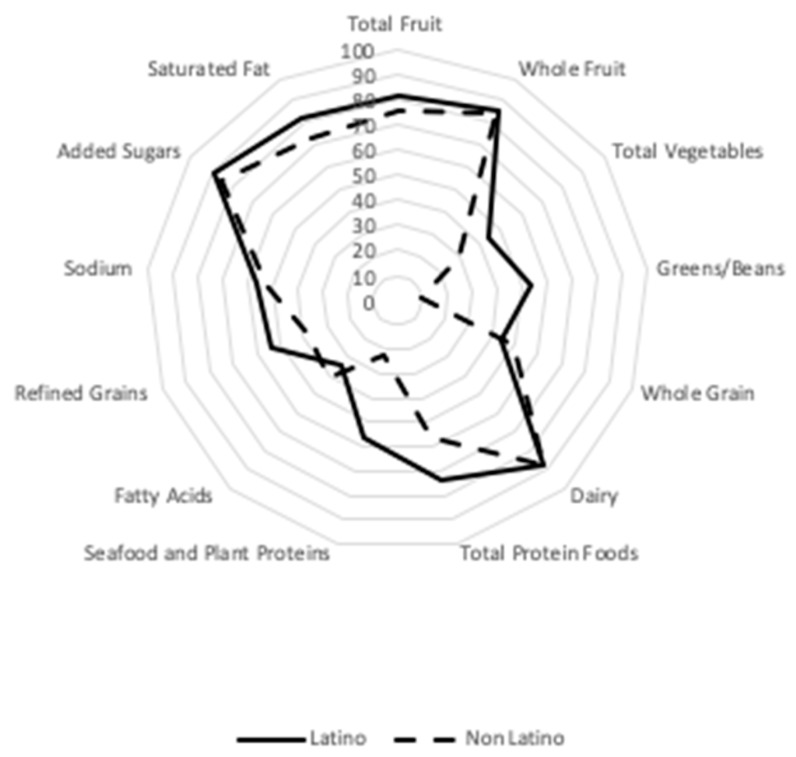
Radar chart of HEI-2015 for 119 FCCH by ethnicity.

**Table 1 nutrients-12-01368-t001:** Demographic Characteristics for 119 family childcare homes (FCCH) by mean Healthy Eating Index (HEI)-2015 scores.

	% (*n* = 119)	Total HEI Score Mean (SD)	*p*-Value
Age (mean, sd)	48.9 (9.0)		-
<50 years	58.3	61.2 (11.4)	0.52
≥50 years	49.2	62.5 (10.6)	
Race			
White	42.5	60.8 (10.8)	0.4
Black	15.0	60.2 (13.7)	
Other	34.2		
Income			
Less than $25,000	13.8	64.8 (11.0)	0.03
$25,001–$50,000	69.8	62.9 (10.8) ^a^	
$75,001 or more	16.4	56.2 (10.6) ^b^	
Marital Status			
Single, never married	9.2	66.6 (10.1)	0.21
Married or living with partner	75.0	60.9 (11.2)	
Divorced/separated/widowed	15.8	63.4 (10.0)	
Education			
High School or GED	43.3	60.7 (10.3)	0.34
Associate/Bachelor/Graduate	56.7	62.7 (11.5)	
CDA (Child Development) Credential			
Yes	27.7	62.9 (11.3)	0.56
No	72.3	61.6 (10.9)	
Ethnicity			
Non-Hispanic	32.5	56.6 (10.0)	0.002
Hispanic	67.5	64.4 (10.6)	
Years in country (mean, sd)			
Nativity			
US-born	29.2	59.2 (11.2)	0.24
<23 years	35.8	63.2 (11.2)	
≥23 years	35.0	62.5 (10.5)	
Language spoken outside of childcare			
English and More English and Equal	45.4	58.6 (10.6)	0.005
Spanish or More Spanish	54.6	64.3 (10.7)	
Language spoken in childcare			
English and More English and Equal	54.2	59.6 (11.3)	0.02
Spanish or More Spanish	45.8	64.4 (10.1)	
Years worked in childcare			
<12 years	48.3	60.7 (10.2)	0.27
≥12 years	51.7	62.9 (11.6)	
Hours worked per week			
Less than 50 h/week	27.5	59.4 (11.0)	0.14
More 50 h/week	72.5	62.7 (10.9)	
Number of children enrolled in childcare			
<6 children	21.0	62.1 (8.3)	0.89
≥6 children	78.9	61.8 (11.7)	
Receives Child Adult Care Food Program Subsidies			
Yes	82.5	62.5 (10.4)	0.14
No	17.5	58.6 (13.1)	

^a,b^ Unlike superscript letters indicate post hoc differences between groups (*p* < 0.05).

**Table 2 nutrients-12-01368-t002:** Regression results for HEI-2015 scores with predictors that were significant in ANOVA.

	Beta Coefficient	SE	*p*-Value
Model 1: Income (Adjusted R^2^: 0.05)			
Less than $25,000	ref		
$25,001–$50,000	−3.2	3.0	0.3
$75,001 or more	−9.8	3.7	0.009
Model 2: Ethnicity (Adjusted R^2^: 0.11)			
Non-Latinx	ref		
Latinx	8.0	2.0	0.0001
Model 3: Language outside childcare (Adjusted R^2^: 0.07)			
English and More English and Equal	ref		
Spanish or More Spanish	5.9	1.9	0.003
Model 4: Language in childcare (Adjusted R^2^: 0.05)			
English and More English and Equal	ref		
Spanish or More Spanish	5.1	2.0	0.01
Model 5: Income, Ethnicity, and CACFP (Adjusted R^2^: 0.12)			
Less than $25,000	ref		
$25,001–$50,000	−2.9	3.0	0.3
$75,001 or more	−5.6	4.0	0.17
Non-Latinx	ref		
Latinx	6.5	2.4	0.007
Childcare does not accept CACFP subsidies	ref		
Childcare accepts CACFP subsidies	4.6	2.6	0.08
Model 6: Income, Language outside childcare, and CACFP (Adjusted R^2^: 0.08)			
Less than $25,000	ref		
$25,001–$50,000	−3.3	3.0	0.27
$75,001 or more	−8.0	4.0	0.05
English and More English and Equal	ref		
Spanish or More Spanish	3.6	2.2	0.09
Childcare does not accept CACFP subsidies	ref		
Childcare accepts CACFP subsidies	3.8	2.6	0.15
Model 7: Income, Language in childcare, and CACFP (Adjusted R^2^: 0.07)			
Less than $25,000	ref		
$25,001–$50,000	−2.5	3.2	0.42
$75,001 or more	−8.1	4.1	0.05
English and More English and Equal	ref		
Spanish or More Spanish	3.1	2.2	0.17
Childcare does not accept CACFP subsidies	ref		
Childcare accepts CACFP subsidies	4.4	2.6	0.1

**Table 3 nutrients-12-01368-t003:** Mean Healthy Eating Index (HEI-2015) total and component scores of foods consumed by children of ages 2 to 5 years attending family childcare homes (*n* = 120) in Rhode Island and Massachusetts by income and ethnicity.

HEI-2015 Component (Maximum Score)	All Children Mean (SD)	Less Than $25,000	$25,001–$50,000	$75,001 or More	*p*-Value	Latinx	Non-Latinx	*p*-Value
Total Fruit (5) ^1^	4.0 (1.0)	4.2 (1.3)	4.0 (1.1)	3.9 (1.3)	0.85	4.1 (1.1)	3.8 (1.2)	0.12
Whole Fruit (5) ^2^	4.2 (1.2)	3.7 (1.9)	4.3 (1.0)	4.2 (1.1)	0.15	4.2 (1.3)	4.2 (1.0)	0.9
Total Vegetables (5) ^3^	2.0 (1.4)	2.1 (1.1)	2.2 (1.5)	1.3 (1.2)	0.06	2.2 (1.4)	1.5 (1.3)	0.02
Greens/Beans (5)	1.9 (2.0)	2.5 (2.1) ^a^	2.3 (2.1) ^a,c^	0.6 (1.0) ^b^	0.004	2.7 (2.0)	0.5 (1.1)	0.0001
Whole Grain (10)	4.5 (3.3)	4.0 (3.6)	4.4 (3.2)	5.3 (3.5)	0.49	4.4 (3.5)	4.9 (2.9)	0.44
Dairy (10) ^4^	8.8 (1.9)	8.7 (2.4)	8.8 (1.8)	8.6 (1.9)	0.88	8.8 (2.0)	8.8 (1.6)	0.9
Total Protein Foods (5) ^5^	3.4 (1.6)	3.9 (1.4) ^a^	3.6 (1.5) ^a,c^	2.6 (1.9) ^b^	0.02	3.7 (1.5)	2.8 (1.7)	0.005
Seafood and Plant Proteins (5) ^5,6^	2.2 (2.1)	3.1 (2.2) ^a^	2.4 (2.1) ^a,c^	0.9 (1.5) ^b^	0.004	2.8 (2.0)	1.1 (1.7)	0.0001
Fatty Acids (10) ^7^	3.6 (2.9)	3.5 (3.3)	3.7 (2.9)	3.3 (2.6)	0.81	3.4 (2.8)	4.1 (3.0)	0.16
Refined Grains (10)	4.9 (3.3)	5.7 (3.2)	5.1 (3.3)	3.9 (3.5)	0.25	5.4 (3.3)	3.8 (3.0)	0.01
Sodium (10)	5.6 (2.9)	6.1 (2.9)	5.3 (2.9)	5.7 (3.2)	0.64	5.7 (2.8)	5.3 (3.0)	0.56
Added Sugars (10)	8.9 (1.8)	8.8 (2.5)	8.8 (1.6)	8.9 (1.7)	0.91	8.9 (1.8)	8.5 (1.9)	0.3
Saturated Fat (10)	7.9 (2.4)	8.6 (2.9)	8.0 (2.3)	6.9 (2.0)	0.10	8.2 (2.5)	7.3 (2.2)	0.05

Food was captured using the Diet Observation at Child Care protocol across two full days. ^a,b,c^ Unlike superscript letters indicate post hoc differences between groups (*p* < 0.05). ^1^ Includes 100% fruit juice. ^2^ Includes all forms except juice. ^3^ Includes legumes (beans and peas). ^4^ Includes all milk products, such as fluid milk, yogurt, cheese, and fortified soy beverages. ^5^ Includes legumes (beans and peas). ^6^ Includes seafood, nuts, seeds, soy products (other than beverages), and legumes (beans and peas). ^7^ Ratio of poly- and monounsaturated fatty acids (PUFAs and MUFAs) to saturated fatty acids (SFAs).

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
