# Peer review of "Exploring the Provider-Level Socio-Demographic Determinants of Diet Quality of Preschool-Aged Children Attending Family Childcare Homes"

_nutrients, 2020, doi:10.3390/nu12051368_

Round 1
Reviewer 1 Report
Thank you for the opportunity to review. Please find my comments attached.

Author Response
Thank you for the invitation to revise and re-submit our manuscript now entitled “Exploring Provider-level Socio-Demographic Determinants of Diet Quality of Preschool Aged Children Attending Family Childcare Homes.” We have revised the paper based on the reviewers’ comments and believe that the manuscript has been strengthened by their suggestions.
Below, please find our point-by-point response to the reviewer comments. Changes to the manuscript are highlighted in light grey.
Section |
Comments |
Response |
Introduction |
· Adequately described |
|
Methods |
· Adequately described
|
|
Results |
· Table 1: What do the subscripts “a” and “b” stand for? · Paragraph after Table 1 is hard to follow. Authors moving between multiple linear regression models and univariate models. Please describe the univariate models first and then move to multiple linear regression models. · Last sentence of paragraph in Page 11- please replace “results” with “differences” · Table 2: Please list the reference category first (e.g., Children does not accept CACFP subsidies” · Figures 1 and 2: Nice way of showing the results. I have not seen Radar charts in the past. However, please use dotted lines or other marks so that differences can be understood in black and white.
|
· We have clarified the subscripts · We have revised the manuscript to first talk about the univariate models and then multivariate models. · We have made the change and replaced “results” with “differences” · Table 2: Have listed the reference category · We have made the graphics more amenable to black and white printing |
Discussion |
· It is contrary to research findings that low-income providers have a higher likelihood of serving nutritious meal. Did the authors adjust for different measures of acculturation (e.g., the duration of years in the U.S. and language spoken at home and country of origin) when examining the relation between income and HEI score? There is some unmeasured confounding going on here. These results need to be replicated in other settings to validate.
|
· We agree with the reviewers that it was somewhat unexpected to find that low-income providers have higher likelihood of serving nutritious meals. In model 5- we include income, ethnicity & CACFP and income is no longer significant in this model but rather ethnicity is. In model 6, when we include income, language outside of childcare & CACFP, the p-value is marginally significant. We hypothesized that this finding would be driven by CACFP status (that perhaps lower income homes would be participating in CACFP which could lead to children consuming healthier meals). However, this was not the case for when we adjust for CACFP in the model, income remained significant. When including ethnicity, language, and CACFP, ethnicity was the only variable that remained significant. It is worth noting however that several of these variables were correlated and might be hard to disentangle. For example, there were a larger % of FCCH that were low-income and Latinx (within the Latinx FCCH: 94% had incomes of Less than $25,000, 74% had incomes of $25,001-$50,000 and 16% had incomes to $75,001 or more). Within, the low-income group, 2.5% where non-Latinx as compared to 19.2% Latinx). Clearly, there may be unmeasured confounding that we are not able to account for. We have added a sentence to the discussion question to address this. It now states: “Second, there may other unmeasured confounding factors which we were unable to adjust for. Future studies should be replicated in other settings to validate our findings.”
|
· Typo: trainings spelling in first paragraph of page 17 “In one of these qualitative studies, Latinx providers discussed the importance of several positive practices including becoming familiar with foods served at home, encouraging healthy foods, role modeling, the importance of meal planning and buying and preparing meals in advance, in addition to participating in nutrition workshops and trainings.” |
· We have fixed the typo |
Reviewer 2 Report
- Authors are suggested to check for grammatical errors such as “healthful children’’ to healthy children and proofread.
- Authors are suggested to provide the difference between total fruit, whole fruit in Table 3. Which fatty acids are there?
- Is Dairy includes all dairy products? Please clarify.
Author Response
Thank you for the invitation to revise and re-submit our manuscript now entitled “Exploring Provider-level Socio-Demographic Determinants of Diet Quality of Preschool Aged Children Attending Family Childcare Homes.” We have revised the paper based on the reviewers’ comments and believe that the manuscript has been strengthened by their suggestions.
Below, please find our point-by-point response to the reviewer comments. Changes to the manuscript are highlighted in light grey.
• Authors are suggested to check for grammatical errors such as “healthful children’’ to healthy children and proofread. • Authors are suggested to provide the difference between total fruit, whole fruit in Table 3. Which fatty acids are there? • Is Dairy includes all dairy products? Please clarify. |
1 Includes 100% fruit juice. 2 Includes all forms except juice. 3 Includes legumes (beans and peas). 4 Includes all milk products, such as fluid milk, yogurt, and cheese, and fortified soy beverages. 5 Includes legumes (beans and peas). 6 Includes seafood, nuts, seeds, soy products (other than beverages), and legumes (beans and peas). 7 Ratio of poly- and monounsaturated fatty acids (PUFAs and MUFAs) to saturated fatty acids (SFAs).
|